# Relating Moisture Transport to Stable Water Vapor Isotopic Variations of Ambient Wintertime along the Western Coast of Korea

**Songyi Kim [1,2], Yeongcheol Han [2], Soon Do Hur [2], Kei Yoshimura [3] and Jeonghoon Lee [1,\*]**

[1]  Department of Science Education, Ewha Womans University, Seoul 03760, Korea; songyikimmm@gmail.com
[2]  Division of Polar Paleo-Environment, Korea Polar Research Institute, Incheon 21990, Korea;
    yhan@kopri.re.kr (Y.H.); sdhur@kopri.re.kr (S.D.H.)
[3]  Institute of Industrial Science, University of Tokyo, Komaba, Tokyo 277-8568, Japan; kei@iis.u-tokyo.ac.jp
\*  Correspondence: jeonghoon.d.lee@gmail.com; Tel.: +82-2-3277-3794

**Abstract:** Atmospheric water vapor transfers energy, causes meteorological phenomena and can be modified by climate change in the western coast region of Korea. In Korea, previous studies have utilized precipitation isotopic compositions in the water cycle for correlations with climate variables, but there are few studies using water vapor isotopes. In this study, water vapor was directly collected by a cryogenic method, analyzed for its isotopic compositions, and used to trace the origin and history of water vapor in the western coastal region of Korea during the winter of 2015/2016. Our analysis of paired mixing ratios with water vapor isotopes can explain the mechanism of water vapor isotopic fractionation and the extent of the mixing of two different air masses. We confirm the correlation between water vapor isotopes and meteorological parameters such as temperature, relative humidity, and specific humidity. The main water vapor in winter was derived from the continental polar region of northern Asia and showed an enrichment of 10 per mil ($\delta^{18}$O) through the evaporation of the Yellow Sea. Our results demonstrate the utility of using ground-based isotope observations as a complementary resource for constraining isotope-enabled Global Circulation Model in future investigations of atmospheric water cycles. These measurements are expected to support climate studies (speleothem) in the west coast region of Korea.

**Keywords:** water vapor isotope; meteorological parameter; mixing ratio

## 1. Introduction

The hydrogen and oxygen isotopic compositions of water vapor can be used as a tracer in the water cycle because these features provide information on water movement, mixing, and phase changes in the atmosphere [1]. Stable water isotopes are characterized by their isotopic compositions which can alter only during physical changes such as evaporation, condensation, and sublimation. Atmospheric water vapor is the source of precipitation, meaning the isotopic compositions of this water vapor directly affect those of precipitation events [2–4]. The isotopic compositions of atmospheric water vapor can provide abundant information on hydrological, ecological, and climatological processes [2,5], and the observation of stable isotopic variations in water vapor is essential to revealing air mass advection, precipitation, evapotranspiration (ET), and entrainment from the free atmosphere [6–11]. In addition, paleo-climate studies using precipitation isotopes require an understanding of water circulation in regions for past climate reconstruction. However, only the measurement of precipitation isotopes has been made in ongoing paleo-climate studies in Korea [12,13].

To express isotopic compositions, delta notation is used to report stable isotopic variations in water samples, i.e.,

$$\delta = (R_{sample}/R_{standard} - 1) \times 1000, R = {}^2H/{}^1H, {}^{18}O/{}^{16}O, \tag{1}$$

which represents the heavy-to-light isotope ratio of the sample ($R_{sample}$) with respect to the reference standard "Vienna Standard Mean Ocean Water" ($R_{standard}$). We calculated the deuterium excess (hereafter *d*-excess), which was defined by Ref. [14], following Equation (3), i.e.,

$$d = \delta D - 8 \times \delta^{18}O \tag{2}$$

A number of studies have been conducted on the tracking of water vapor sources with precipitation isotope ratio and local weather and spatial variables (e.g., temperature, relative humidity, latitude, altitude, precipitation amount and so on) [15–19]. However, little has been reported on isotope distributions in the water vapor phase around the Korean peninsula because of limitations in instruments and techniques for atmospheric water vapor isotopic measurement.

Winter over the Korean peninsula is represented by Siberian high pressure and the strength of the upper-tier jet and the tectonic bore in East Asia under the influence of the East Asian monsoon circulation. In particular, the intensity of Siberian high pressure is closely related to the winter cold wave within the Korean peninsula. During the winter of 2015/2016, the effect of high-latitude climate variability was expected to have been complicated by a strongly developed El Niño and Arctic sea ice extent [20]. In the Korea peninsula, precipitation isotopic compositions alone were used to comprehend the characteristics of moisture [21,22]. Ref. [21] describes the precipitation isotope ratio on the Korean peninsula as being determined by a humid South Pacific air mass in summer which is affected by the amount effect. In winter, a dry polar air mass passes through the Asian continent and collects water moisture through the Yellow Sea. The water vapor isotopes during periods of no precipitation can be used to interpret the characteristics of the moisture source. Two theoretical models have been used for this purpose. First, Rayleigh distillation theory assumes the water vapor isotopic ratio at a certain air mass decreases as the condensation temperature and saturated specific humidity decrease [1,23–25].

$$\delta - \delta_0 \approx \ln(R/R_0) = (\alpha(T) - 1) \times \ln(q/q_0) \tag{3}$$

At equilibrium, the surface seawater temperature determines the isotopic composition of the resultant water vapor as given in Ref. [6].

where $\delta_0$ and $q_0$ are the sea water isotopic composition and the saturated water amount, respectively, $\delta$ and q are the final water vapor isotopic composition and the final water vapor mixing ratio, respectively, and T and $\alpha$ are the sea surface temperature (SST) at evaporation and the temperature-dependent equilibrium fractionation factor between vapor and condensates, respectively [26]. This model assumes that isotope fractionation occurs as water vapor condenses in an adiabatic system [1]. Hence, each time condensation occurs, the precipitation and water vapor isotopic compositions are depleted. Secondly, a mixing model has been used to analyze the humidity and isotopes under various conditions [27]. This mixing model for isotopic compositions is

$$\delta D x q = \delta D_F \times (q - q_0) + \delta D_0 + q_0 \tag{4}$$

where $\delta$ is the isotopic composition and q is the water mixing ratio. The subscript *F* denotes the flux into the volume of interest and the subscript 0 is the initial flux. The $\delta_F$ was calculated as the slope of Equation (6) in Ref. [8].

In this work we aimed to investigate the moisture transport mechanisms in the wintertime in the Korean peninsula using water vapor isotopes. To accomplish this, we collected water vapor by a cryogenic method, and the analyzed water vapor isotopes were correlated with meteorological variables. The isotopic values of water vapor were evaluated by model results. This outline of this paper is laid out as follows. In Section 2 of this paper we present the study area and discuss the

methodology. In Section 3, we document that our measurements are typically for the winter season at mid-latitudes. The moisture source as investigated from back-trajectory results and the Rayleigh distillation model and mixing model are reported. Finally, our conclusions are drawn in Section 4.

## 2. Methods

### 2.1. Study Area

Samples of atmospheric water vapor were collected between 3 December 2015 and 28 February 2016 at the Korea Polar Research Institute (KOPRI) in Incheon (Figure 1) in the western part of South Korea (37°22′03 N 126°38′47 E, 0 m a.s.l.). The sampling point was placed on the 5th floor of a six-story building at KOPRI (around 10 m a.s.l.), which is approximately 500 m away from the western coast of South Korea. The sampling area, which has been reclaimed, was chosen because there was no forestry around, meaning the water vapor isotopic ratio from transpiration was thought to be minimal.

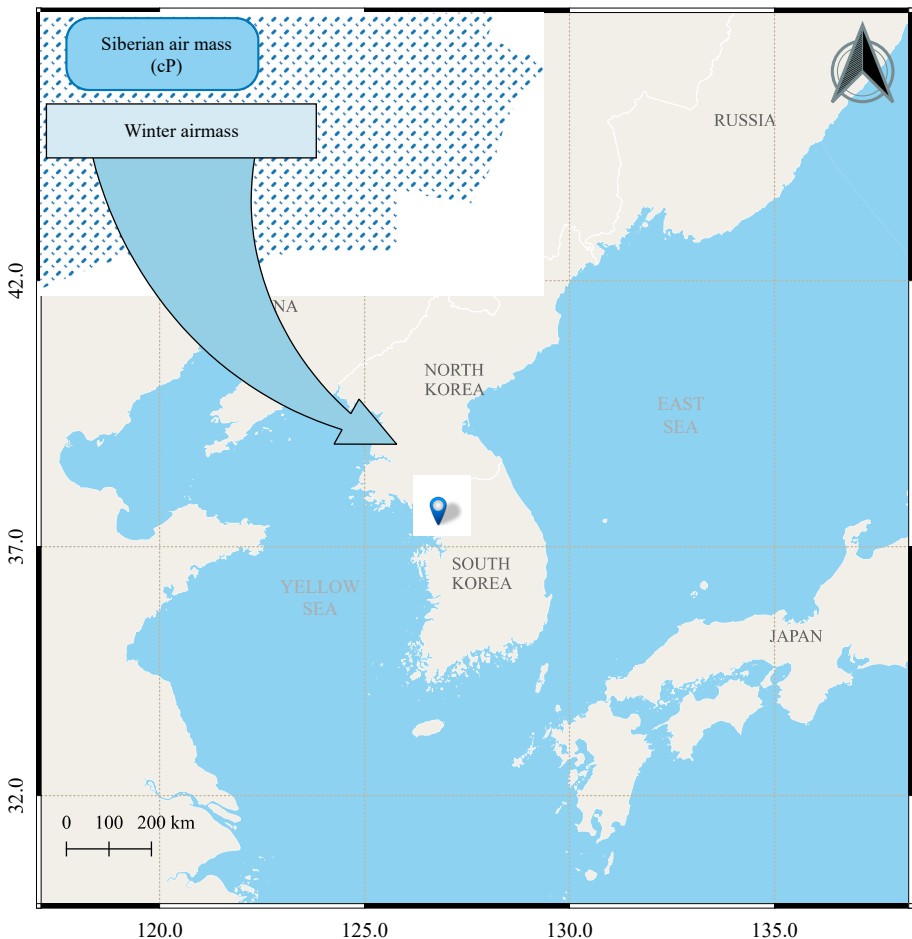

**Figure 1.** The study area around the Korean peninsula.

The Korean peninsula, which is located in East Asia, is classified as a temperate climatic zone with four distinct seasons. Generally, a prevailing process of the mid-latitude zones over East Asia in winter is the invasion of extremely dry and cold air over Mongolia, northern China, and Korea. This strong anticyclone pressure, which is centered over continental Asia, controls almost the entire region of Eurasia [28]. The winter weather in 2015/2016 appeared to be a strong cold wave which occurred in mid-January [20]. Meteorological data from our study period, which were recorded by an automatic weather station in the vicinity of the sampling location (approximately 10 km away from the sampling location at 37°28′39 N 126°37′29 E, about 70 m a.s.l.), were used. In particular,

an analysis of the water-isotopic composition was not possible because the amount of precipitation was insufficient for measurement during the sampling period (i.e., it was less than 1 mm each day).

## 2.2. Sample and Data Acquisition

We used a cryogenic sampling method which collects condensed water vapor in the form of both ice and liquid water. This sampling method has been detailed in a previous study [29]. To minimize the isotopic fractionation when sampling water vapor, these authors tested the method by using a serial connection with two sets of impinger devices. Considering the amount of trapped water vapor in the two impingers, the isotopic fractionation between the two impingers (0.06‰ for oxygen and 0.33‰ for hydrogen, respectively) was less than or equal to analytical errors (0.06 and 0.7 for oxygen and hydrogen, respectively). Thus, the amount of water vapor that was trapped and the isotopic fractionation in the second impinger were negligible.

The arms of the impingers, which were connected to flexible tubing through an air-sampling pump, were placed to allow a connection to ambient air. Dewer bottles that contained these impingers were filled with liquid nitrogen at low temperature ($-196\ ^\circ$C) to prevent further isotopic fractionation from incomplete sampling. Typically, each sampling duration was 6 h with a flow rate of 1 L/min (one or two samples per day). Each impinger, in which water vapor condensed to ice or liquid water was held, was sealed with a parafilm and put in a refrigerator as any ice melted. After melting, the water was filtered and injected into vials and kept in a refrigerator before analysis. We acquired fifty-four condensed vapor samples during the study period.

Meteorological conditions such as air temperature, relative humidity, and vapor pressure were obtained using an automated weather station (AWS). We used an hourly averaged meteorological dataset which was based on the Automated Synoptic Observing System (ASOS) (Korean Meteorological Administration, https://data.kma.go.kr/). Hourly data were averaged over the water vapor sampling interval to observe the relationships between the meteorological parameters and water vapor isotopic compositions. Moreover, the water vapor pressure and the atmospheric pressure were used to calculate a mixing ratio using an following equation from Ref. [30], i.e.,

$$w = \varepsilon \times e/(P - e) \tag{5}$$

where $w$ is the actual water vapor dry mass mixing ratio, $\varepsilon$ is the ratio of the molecular weight of water and dry air ($\approx 621.9907$), $e$ is the water vapor pressure, and $P$ is the ambient pressure.

## 2.3. Isotope Analysis

The samples were analyzed simultaneously for $\delta^{18}$O and $\delta$D by using a cavity ring-down spectroscope (CRDS, manufactured by Picarro, Inc., Santa Clara, USA) at the KOPRI. One batch consisted of 10 samples and one laboratory-made standard (Styx). Before several batch analyses, three reference standards, namely, Vienna Standard Mean Ocean Water (VSMOW2), Greenland Ice Sheet Precipitation (GISP), and Standard Light Antarctic Precipitation (SLAP2), were measured for calibration. Each sample and reference standard was injected 12 and 20 times, respectively. To avoid any memory effects from previous samples, the last six injections of each sample were used. The precision of the isotopic analysis was 0.06‰ and 0.7‰ for oxygen and hydrogen, respectively.

## 2.4. Air Mass Trajectory Analysis

The unrealized locations of moisture sources and traces of air mass before reaching the study site were estimated using back-trajectory analysis [31,32]. This estimation was executed using the Hybrid Single-Particle Lagrangian Integrated Trajectory (HYSPLIT) model, which was developed by the National Oceanic and Atmospheric Administration (NOAA)'s Air Resources Laboratory [33,34]. The HYSPLIT model's calculation method is a hybrid between a Lagrangian approach and Eulerian methodology. a $1^\circ \times 1^\circ$ climate dataset was provided by the global data-assimilation system (GDAS)

for the depiction of airflow patterns. Four-day back-tracking analysis was performed every 1 h from 1 December 2015 to 28 February 2016 at the sampling site for 100 m above ground level. Then, the output results were segregated for comparison with our observations.

## 3. Results and Discussion

### 3.1. Temporal Variations in the Water Vapor Isotopic Composition

Atmospheric vapor isotopic values ($\delta^{18}$O, $\delta$D, and *d*-excess) with AWS observations of air temperature, relative humidity, and mixing ratio during the study period were plotted (Figure 2). Air temperature and relative humidity ranged from −12.5 °C to 14.5 °C (0.8 °C of the mean temperature) and from 35 to 98% (62.3% of the mean relative humidity), respectively. The total precipitation during the 2015/2016 winter season was 77.9 mm, which was higher than the normal annual precipitation (60.7 mm during the period 1981–2010). Precipitation sampling in the winter of 2015/2016 was difficult because of evaporation. The sum of winter precipitation during the sampling period was 5.05 mm. The $\delta^{18}$O value of water vapor ranged from −34.04‰ to −15.27‰, with a mean value of −24.69‰, and the $\delta$D value ranged from −215.8‰ to −100.2‰, with a mean value of −153.5‰. The mean value of *d*-excess for the water vapor samples was 44.0‰ (between 17.4‰ and 68.3‰).

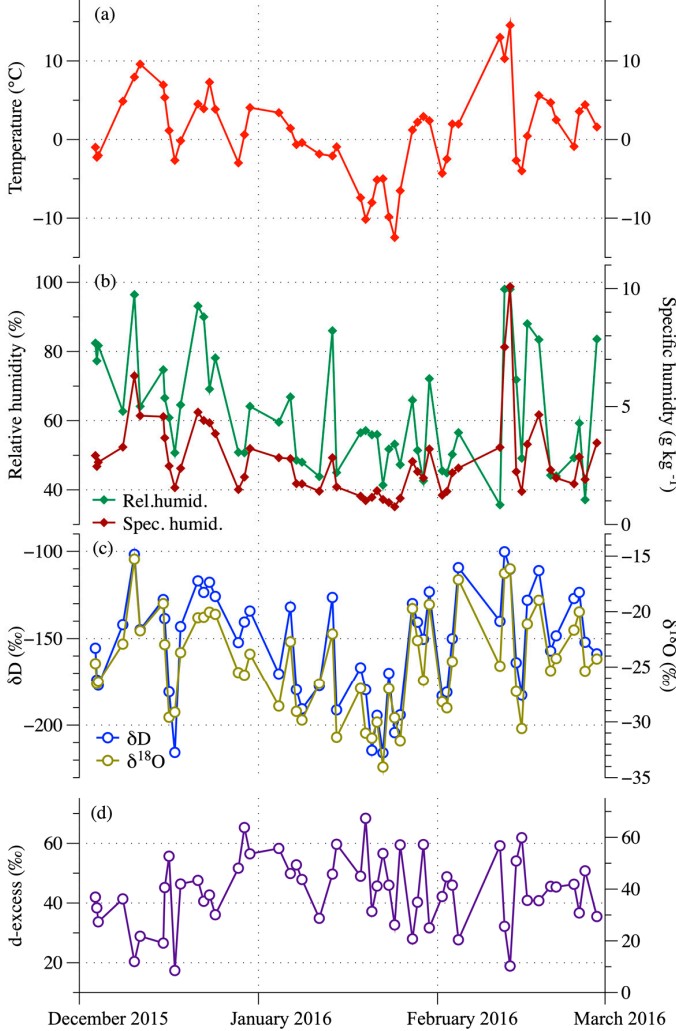

**Figure 2.** Time series of (**a**) air temperature, (**b**) relative humidity and mixing ratio, (**c**) stable oxygen and hydrogen isotopic compositions of water vapor, and (**d**) *d*-excess.

Variability of precipitation isotopic compositions has been observed in Korea, but only for the rainy period [21,22]. Previous studies have reported that the dry winter air that passes through the continent collects moisture as it passes through the Yellow Sea, resulting in precipitation in winter over the Korean peninsula. In our study, we discuss the role of air sources and mixing during the non-rainy season and its effect on the isotopic values of water vapor. The results from these previous works show that depleted precipitation isotopic compositions in summer occur because of an isotope-amount effect, while enriched precipitation isotopic compositions in winter are caused by evaporated water vapor in the Yellow Sea. As a result, the seasonal pattern of temperature in Korea and the precipitation isotope pattern are different.

However, our study shows that temperature and water vapor isotopic compositions are covariant, demonstrating variability in water vapor isotopes which is similar to that of temperature. Water vapor isotopic composition could be determined under the influence of temperature, so the two relationships were correlated. During warm periods, the range of water vapor isotopes ($\delta^{18}O$) was −24.92‰ to −15.27‰, with the mean value being −19.88‰. The water isotopic values of oxygen in the cold period were −34.04‰ to −26.97‰, with the average value being −30.0‰ (an almost 10‰ difference). In addition, the temperature range was 4.9 °C to 14.5 °C during warm periods and −12.5 °C to −4.0 °C during cold periods. During the wintertime, the isotopic variability by temperature between the two periods (warm versus cold) was distinctive.

At the time of the cold wave, the isotopic values of water vapor were depleted, suggesting that water vapor isotopes act as an indicator of temperature in the wintertime over the Korean peninsula. As with temperature, absolute humidity exhibited similar variability to the isotopic values of water vapor, indicating the presence of a low amount of water vapor during the cold-surge season. This observation might have been the result of isotopic fractionation. The correlation between temperature and water vapor isotopic composition will be discussed in detail in the next section.

### 3.2. Interpretations of the Relationship between Climate Variables and Water Vapor Isotopes

In previous studies, the isotopic compositions of ambient water vapor have been related to climate parameters such as local air temperature and atmospheric humidity [8,24,31,35–37]. Ref. [35] found that the monthly mean water vapor isotope values ($\delta18O$ and $\delta D$) were positively correlated with monthly mean temperature, producing linear correlation coefficients of 0.87 and 0.88, respectively. This temperature effect, which has been observed in precipitation isotopic compositions in high latitude regions, could also be found in water vapor isotopic compositions [1,8]. However, Ref. [36] has argued that atmospheric humidity is a better predictor of water vapor isotopes than air temperature. Covariations in water vapor isotopic ratios with regional climatic factors in the study area, i.e., temperature, relative humidity, and mixing ratio, were estimated (see Figure 3 and Table 1). Regression-analysis modeling indicated a significant linear correlation between $\delta$ values for water vapor and air temperature which comprised roughly 50% of the variance (Figure 3a). The $\delta18O$ and $\delta D$ values with water vapor mixing ratio relationships were positive log-linear, and the correlation coefficients for these data were 0.68 and 0.64, respectively (see Table 1 for details). Additionally, positive relationships between the isotopic values of water vapor and relative humidity can be observed ($R2 = 0.42$ and 0.34 for oxygen and hydrogen, respectively). These results enabled us to compare the values of water vapor isotopes with the Rayleigh distillation model (see Equation (3)), with the remaining water vapor amount, water vapor isotope values, and condensate–precipitation isotope values determined from the condensation temperature in clouds. In the following section, we discuss how water vapor was affected in our study by using models that consider water vapor isotopes and climate variables.

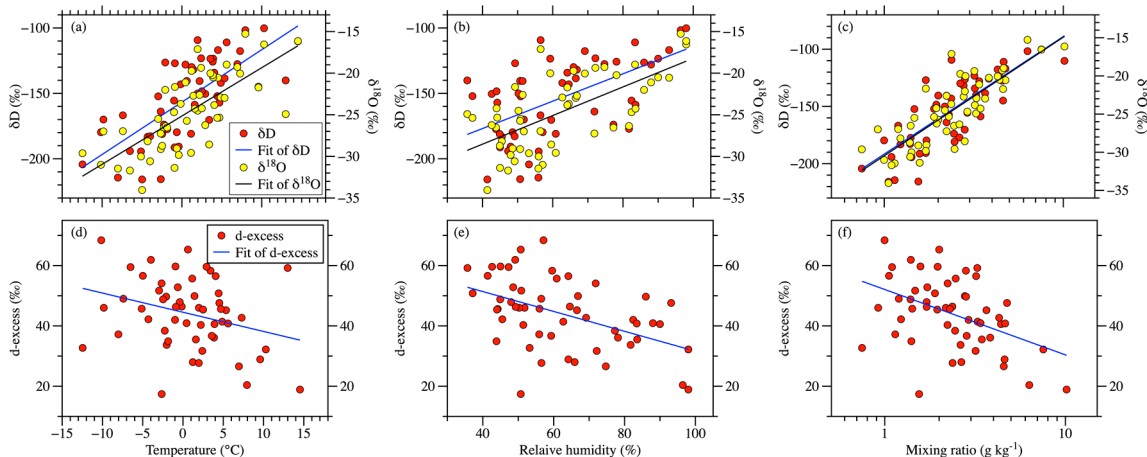

**Figure 3.** Scatter plot and linear and log-scale regressions of isotopic values and meteorological parameters: $\delta^{18}O$ and $\delta D$ with (**a**) temperature, (**b**) relative humidity, and (**c**) mixing ratio for log-scale, and $d$ values with (**d**) temperature, (**e**) relative humidity, and (**f**) mixing ratio for log-scale.

**Table 1.** Linear regression analysis of the relationship between water isotopic compositions and meteorological parameters.

| Line | $\delta^{18}O$ | | | $\delta D$ | | | $d$-Excess | | |
|---|---|---|---|---|---|---|---|---|---|
| | Slope | $R^2$ | $p$ Value | Slope | $R^2$ | $p$ Value | Slope | $R^2$ | $p$ Value |
| *Entire period* | | | | | | | | | |
| Temperature | 0.58 | 0.51 | <0.0001 | 4.0 | 0.52 | <0.0001 | −0.6 | 0.08 | 0.0281 |
| Relative humidity | 0.17 | 0.42 | <0.0001 | 1.0 | 0.34 | <0.0001 | −0.3 | 0.22 | 0.0003 |
| Mixing ratio [†] | 6.75 | 0.68 | <0.0001 | 44.6 | 0.64 | <0.0001 | −9.4 | 0.19 | 0.0009 |
| *Warm period (4.9 °C ~ 14.5 °C)* | | | | | | | | | |
| Temperature | 0.19 | 0.04 | 0.5901 | 0.9 | 0.03 | 0.6293 | −0.6 | 0.03 | 0.6511 |
| Relative humidity | 0.16 | 0.91 | <0.0001 | 0.7 | 0.72 | 0.0018 | −0.5 | 0.65 | 0.0049 |
| Mixing ratio [†] | 7.79 | 0.78 | 0.0007 | 35.9 | 0.58 | 0.0098 | −26.4 | 0.61 | 0.0077 |
| *Cold period (−12.5 °C ~ −4.0 °C)* | | | | | | | | | |
| Temperature | −0.18 | 0.05 | 0.5194 | −0.3 | 0.00 | 0.9043 | 1.2 | 0.09 | 0.4097 |
| Relative humidity | 0.16 | 0.16 | 0.2532 | 0.9 | 0.09 | 0.4032 | −0.4 | 0.03 | 0.6301 |
| Mixing ratio [†] | −0.46 | 0.00 | <0.0001 | 11.6 | 0.02 | 0.0003 | 16.7 | 0.08 | <0.0001 |

[†] Logarithmic scale

The correlation between *d*-excess and meteorological variables was unclear, with low $R^2$ values observed. In particular, for water vapor isotope values that were measured above the sea, the relative humidity and *d*-excess values were strongly correlated, because the *d*-excess values (e.g., the initial water vapor isotopes' relationship evaporated in the ocean) were determined from the kinetic isotope fractionation, which depends on relative humidity. Thus, we could infer from the inappreciable correlation of *d*-excess that no water vapor passed directly through the Yellow Sea. The changes in the initial vapor isotopic composition that formed above the sea could have affected the coastal *d*-excess and must be considered. This effect will be examined alongside the mixing effect in the next section.

*3.3. Physical Processes Affecting Water Vapor Isotopes*

Given that we have related climatic variables to the isotopic compositions of water vapor, we now investigate the origin and transport of water vapor using both Lagrangian and Eulerian approaches.

To deduce processes of isotopic variation other than water vapor directly from the sea, we used a Rayleigh distillation model (see Equation (3)) and mixing model (see Equation (4)). First, we evaluated the water vapor $\delta$D by invoking the equilibrium fractionation effect and compared the model results with the observed $\delta$D values [8]. The slopes of $\delta^{18}$O with temperature were found to be 0.58‰/°C and 0.76‰/°C for our data and a Rayleigh rainout scenario (when the temperature was assumed to be between 0 °C and 25 °C), respectively. This difference in slope suggests that these air masses mixed along air-mass trajectories.

The highest correlations between the weather variables and the isotopic values of water vapor were the relationships of the mixing ratio at log-scale and the observed variations in $\delta^{18}$O and $\delta$D ($R^2$ = 0.68 and 0.64, Table 1). These relationships have proven useful at elucidating mixing processes [1,27]. The amount of water vapor changes when changing water vapor isotopic values (evaporation and condensation, etc.). According to Equation (2), $q_0$ and $\delta_0$ were improvised from ensemble low values among the data that we obtained. In Figure 4, the gray line used the slope of the regression line of the measured q×$\delta$D versus q as the first end member, and the second end member was calculated as −220‰, with 0.7 g kg$^{-1}$. The mixing curves, except for the gray line, were calculated as the first end member of the water vapor isotopes that were obtained from the SST of the Yellow Sea, and the second end members were the water vapor isotopes of −210‰, −220‰, and −230‰, with a mixing ratio of 0.7 g kg$^{-1}$. The second end members in this paper were the ensemble values of the lowest values of the measured parameters. In Figure 4, the observed data lay between the Rayleigh curve and mixing curves, assuming an evaporative source from the ocean. Figure 4 shows the vapor-isotope ratio for the cold-wave period at lower mixing ratios (i.e., around −210‰ to around −230‰), which aligned with the mixing line. Thus, the moisture source of the precipitation isotopic composition in the Korean peninsula was the Yellow Sea [21] when the water vapor isotope values were enriched. During the cold-wave season, the depleted isotopic composition and low amount of water vapor suggest moisture transport from higher latitude than that of the Yellow Sea, meaning the air mass did not pass through the ocean or otherwise was not influenced by the Yellow Sea during transport.

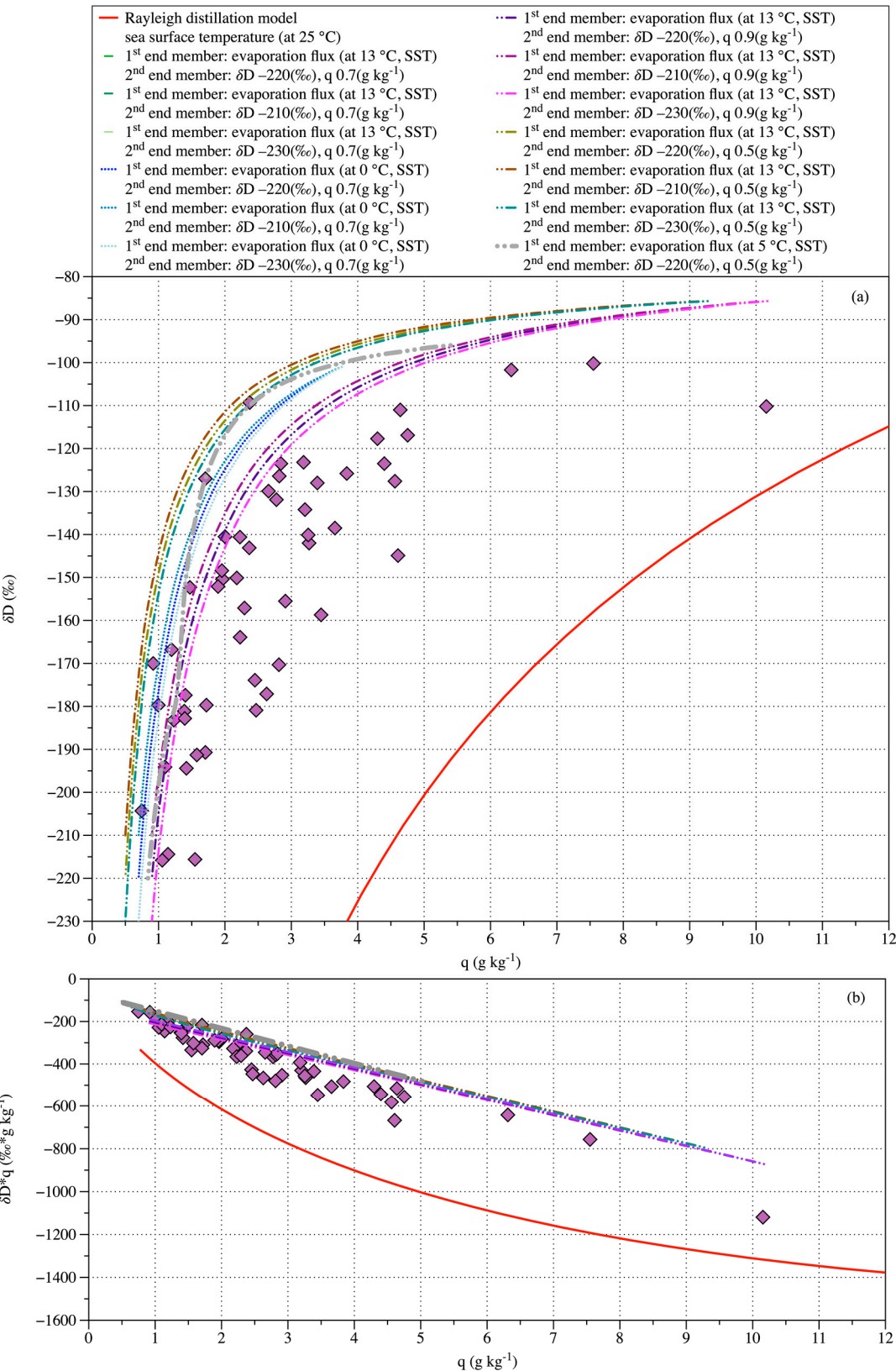

**Figure 4.** (**a**) δD versus water mixing ratio (q) and (**b**) q×δD versus q. The purple diamonds are measurements. The curves are for Rayleigh processes (red solid lines), which assume a source at 100% humidity with a sea-surface temperature of 25 °C, and the mixing model (except for the red solid lines), which reasonably assumes two end members. The conditions for each end member are described in the legend. The slope and intercept of the q×δD/q linear relationship in observation are −95.2 and −123.3, respectively.

### 3.4. Moisture Transport around the Western Korean Peninsula

Patterns of water vapor isotopes are related to air masses, i.e., large bodies of air with nearly uniform temperature and humidity characteristics. These meteorological variables can be influenced by the water vapor's origin, meaning the isotopic components of water vapor could also be affected by the origin of water vapor. In this study, the relatively depleted water vapor isotopic values and low temperature and humidity during the winter cold-surge season over the Korean peninsula indicated a difference in the source of water vapor. Additionally, enriched isotope values and relatively high temperatures and humidity were presumably caused by evaporation from the ocean. By using the relationship between meteorological variables and water vapor isotopes, we first attempt to perceive changes in the water vapor's origin and then describe the effect of the sea's influence during water vapor transport.

In addition, back-trajectory analysis was used in prior research alongside precipitation and water vapor isotope ratios to investigate the sources of moisture [9,21,31,32]. To help interpret the variability in water vapor isotopic compositions, back trajectories for air parcels at the sampling site and time were performed using HYSPLIT. The moisture sources during the study period were monitored and traced to explain the reason for the isotopic differences in Incheon during the winter season.

Figure 5a shows the trajectories when considering the entire measurement time. Each period of high temperature and low temperature (cold-wave period) is shown in Figure 5b,c. During cold surges, the trajectory model shows that the beginning of the trajectory was near Siberia, passing straight towards the coastline of the Korean peninsula through North Asia (Figure 5b). The air parcels flowed directly without passing through other locations, meaning these parcels were not likely influenced by evaporation flux. However, Figure 5c shows air masses that originated over the Chinese continent and mixed with marine air masses (the Yellow Sea). In particular, the water vapor isotope ratio was depleted and the water mixing ratio was lower. We are able to confirm that the sources and pathways of these air masses were different during cold and warm periods.

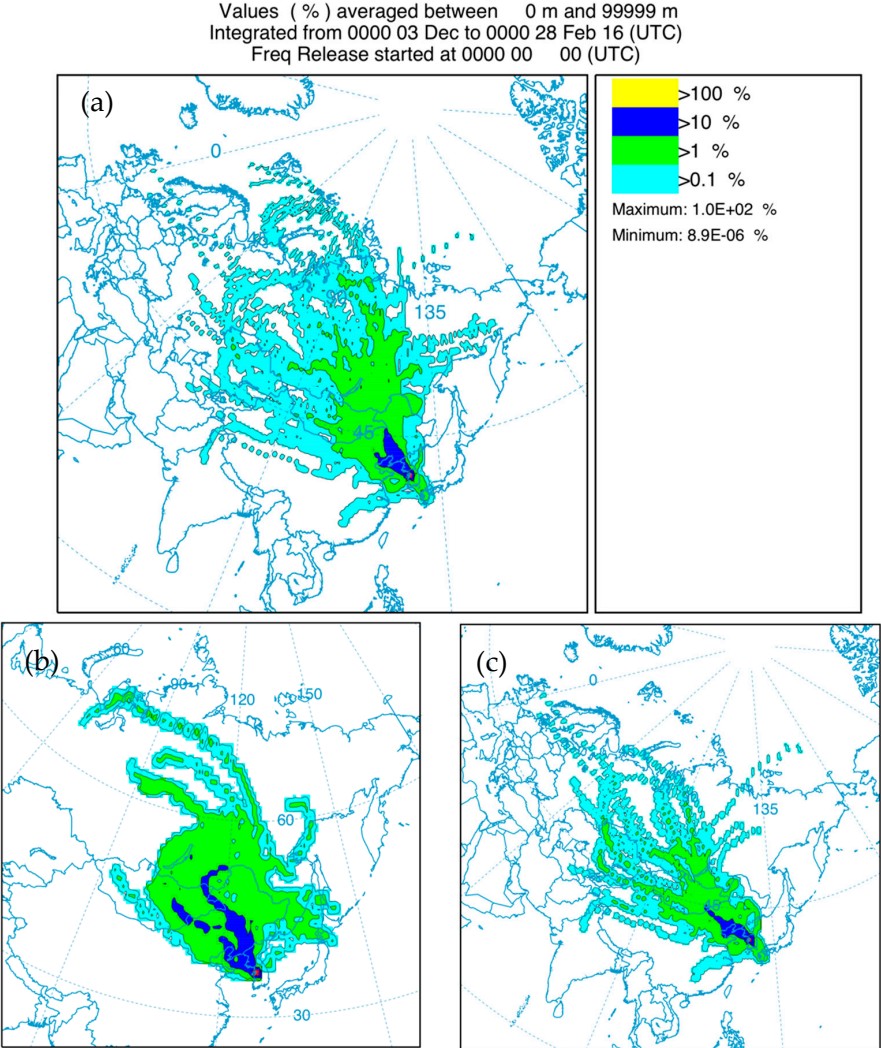

**Figure 5.** (**a**) Hybrid Single-Particle Lagrangian Integrated Trajectory (HYSPLIT) back trajectories showing the locations of wintertime (DJF) back-calculated air parcels 120 h prior to their arrival at the study sites, (**b**) HYSPLIT back trajectories under sub-zero temperature conditions, as shown in Figure 4a (these 40 back trajectories follow a roughly parallel transect across central China), and (**c**) 120-h air-mass back trajectories from the sampling locations with air parcels along the trajectories under lower temperatures (roughly 40 samples).

### 3.5. Comparison between the Observation Results with Isotope-Enabled Global Spectral Model (Iso-GSM) Results

Isotope-enabled atmospheric general circulation models are useful for supplying the global water isotope field. The Iso-GSM incorporates both kinetic and equilibrium fractionation when determining isotopic ratios in vapor after considering the complex atmospheric processes via the spectral nudging technique [38]. Previous studies have been comparative studies between observed and model data [39–41]. Here, we compare our observations with the Iso-GSM. Six-hourly data were generated in the Iso-GSM model corresponding to the times 0, 6, 12, and 18 h. In addition, this model assumes no isotopic fractionation of evapotranspiration fluxes in land surface [38].

To compare our results with this model, four points near the study area were used for the estimated study area by using inverse distance weighting interpolation (Figure 6). Iso-GSM captured most characteristics of the water vapor isotope during the wintertime. The water vapor isotopic values of the Iso-GSM model simulations reproduce well the drastic fluctuations under the winter monsoon. However, the water vapor isotopic values of the Iso-GSM model were more enriched than the observed

ones, which might have resulted from overestimating the evaporation of the ocean. One of the four outcomes in the model simulation was from the Yellow Sea, and two were simulated near the coast, meaning the evaporated convection might have been affected. Additionally, it should be noted that there is an inherent limitation in these comparisons because of the different spatial scales. The results of the model were simulated at 2 m and the study was conducted at a height of about 10 m, which causes some differences. Hence, detailed observations are required for a more accurate interpretation.

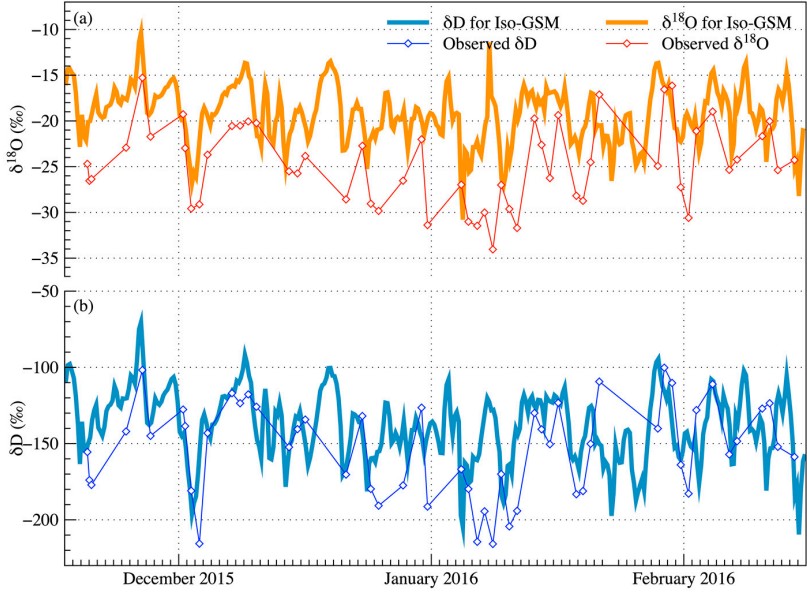

**Figure 6.** Comparison between observed water vapor for (**a**) $\delta^{18}O$ and (**b**) $\delta D$ with Iso-GSM data during the sampling period. The isotope-enabled global spectral model (Iso-GSM) data were calculated by interpolating four different areas near the study area.

## 4. Conclusions

Using isotope observations, we have demonstrated in this work that atmospheric mixing between dry continental air and moist marine air masses controls the near-surface air humidity in the coastal region of Korea. Using this information, we suggest that the transport of marine vapor by advection from the Yellow Sea represents a major source affecting warming in Korea during the wintertime. Our results establish the capacity of using single point, ground-based isotope observations in the study of atmospheric moisture transport. This study combines isotope-enabled GCM modeling with a robust isotope data set to assess model validation. Models tend to over-estimate values of evaporation at sea. a problem might arise from the point of dealing with the spatial scale of the Iso-GSM model.

The results of this study regarding the formation of water vapor isotopes, which are the source of precipitation, can provide information on the water cycle in the wintertime over the Korean peninsula. Hydrological circulation over the Korean peninsula has been studied with precipitation isotopes, but this paper contributes to the study of the water cycle during the non-rainy season. To better understand the water cycle, studies on the relationship between water vapor and precipitation are also needed. By understanding moisture transport in Korean peninsula, we believe this work can increase the accuracy of paleoclimate studies using speleothem.

**Author Contributions:** S.K. was responsible for the implementation of the data collection, processing and writing of the manuscript. J.L. was responsible for the designing and analyzing of the work. S.D.H. and Y.H. provided constructive comments and funding. K.Y. provided the Iso-GCM results and comments of the work.

**Funding:** This research was funded by KOPRI research grants (PE19040) and the Basic Research Program through the National Foundation of Korea (NRF), which was funded by the Ministry of Education (NRF-2017R1D1A1A09000732).

**Conflicts of Interest:** The authors declare no conflict of interest.

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
