# Peer review of "Relating Moisture Transport to Stable Water Vapor Isotopic Variations of Ambient Wintertime along the Western Coast of Korea"

_atmosphere, doi:10.3390/atmos10120806_

Round 1

Reviewer 1 Report

The manuscript is well-conceived and written. This study is worthy of publication in Atmosphere emphasizing the regional importance of climate studies in the west coast region. However, the manuscript is more descriptive than interpretive, rendering it of less interest. 

1) Abstract. The abstract needs to clearly define the questions posed and how the isotopic composition of water vapor addresses the questions. For example, how will the water vapor isotope ratios improve global circulation models? What new climate-related insights are possible for Korean west coast? What role does the Yellow Sea play in climate change effects for the peninsula?

Suggested revision for abstract:

Atmospheric water vapor transfers energy, causes meteorological phenomena and may be altered under climate change in the west coast region of Korea. Previous studies have described precipitation isotopes in the water cycle for correlation with climate variables, but there are few studies using water vapor isotopes. Water vapor isotopes are important for??????????  In this study, water vapor was directly collected using a cryogenic method, analyzed for δ18O and δD  isotopic composition (wintertime over 2015/16) and used to trace the origin and history of the water vapor in the west coastal region of Korea. Analysis of paired mixing ratio with water vapor isotopes explain the mechanism of water vapor isotopic fractionation and the extent of the mixing of two different air masses. We confirm the correlation between the water vapor isotopes and the meteorological parameters including ?????. The main water δ18O vapor in winter was derived from continental Polar of north Asia and showed enrichment of ????? per mil through the evaporation of the Yellow Sea. These measurements are expected to improve the climate models such as global circulation model by ????? and supporting climate studies in the west coast region of Korea.

Revisions to sentence structure and grammar were too numerous to address. I suggest that an editor with native English proficiency read through the manuscript.

Please clarify:

62This model assumes that water vapor dries (do you mean evaporates?)

63 and the water vapor isotopes are reduced ( do you mean become isotopically enriched?) by condensation, which is known to be related to

64 temperature at the time of condensation, vapor isotopes and precipitation.

Please clarify:

70-74 What questions or applications are being addressed? Simply reporting isotopic data has merit but the authors should develop research questions and answers with the data.

Note:

99 less than or equal to analytical errors.

154-161 The isotopic composition of water vapor is expected to co-vary with temperature. I am not clear on the distinction being made between “a reversed change….” and “…so the two relationships are correlated.”

Please clarify:

Are water isotopic data for the Yellow Sea available? An argument is made that the Yellow Sea is a source region but other than a back trajectory, being a very indirect approach, there is no evidence provided to test the question.

Figure 1. Are SD’s  available for delta D and delta 18O?

Figure 3. (a) δD versus water mixing ratio (q) and (b) q δD versus q. [Note, “delta D*q” correction in the caption and (q), parentheses are missing.

Figure 4.  It is very difficult to understand what these back trajectories mean as there is no estimate of uncertainty nor of relating isotopic data to support the simulation exercise. This figure could be eliminated with no loss of interpretation.

Figure 5 and interpretation are far more convincing than Figure 4 and could be amplified and discussed in more detail including a more in-depth interpretation of the published literature in this area.

4. Conclusions

Much of the narrative in the conclusion section should go in the discussion section. The conclusion section might amplify what the study has described that is new and its significance to the science and study of climate change in the region.

Reviewer 2 Report

see file

Author Response

Reply to the comments by reviewer 2

General comment: The manuscript atmosphere-645291 reports a dataset of water vapor isotopes. In this study, water vapor was directly collected using a cryogenic method and isotope compositions were measured to trace the origin and history of the water vapor. Although the subject is well known in the literature, I believe this manuscript should be published after a major revision.

Answer: Thank you for the positive comments on our manuscript.

P2/R40 Please add the paper proposed by Vespasiano et al., 2015 and Apollaro et al. 2019.

Vespasiano G., Apollaro C., De Rosa R., Muto F., Larosa S., Fiebig J., Mulch A., Marini L. (2015): The Small Spring Method (SSM) for the definition of stable isotope - elevation relationships in Northern Calabria (Southern Italy). Applied Geochemistry. 63, 333-346. Apollaro, C., Tripodi, V., Vespasiano, G., De Rosa, R., Dotsika, E., Fuoco, I., Critelli, S. & Muto, F. (2019). Chemical, isotopic and geotectonic relations of the warm and cold waters of the Galatro and Antonimina thermal areas, southern Calabria, Italy. Marine and Petroleum Geology.109, 469-483.

Answer: The papers are now cited in the manuscript. Those papers are helpful in applications of our work.   

P2/Study area - Please, add a general morphological map showing the main reliefs and the direction of the main air masses

Answer: We somewhat agree with the reviewer’s point. Since we were not able to obtain Digital Elevation Map around Korean Peninsula, we add a map showing the direction of air masses in winter. P4/Results and Discussion - it could be useful add a d2H vs. d18O graph with all sampling points compared with the global meteoric water line (Craig, 1961)

Answer: We somewhat agree with what this reviewer pointed out. However, the meteoric water line only tell us what the water has experienced right after the water vapor evaporated from the ocean. Our data shows that the water vapor during our study period experienced some evaporation since the slope of oxygen and hydrogen isotopes was 6.5, which is deviated from the slope of 8 (global meteoric water line). We put the plot of relationship between two water isotopes below.

Fig. Linear relationship between oxygen and hydrogen isotopes for the water vapor

English needs to be reviewed by a native speaker.

Answer: We changed the manuscript for English correction as the reviewer suggested.

Reviewer 3 Report

I attached my comment below.

Author Response

Reply to the comments by reviewer 3

General comments: This paper examines the sources of water vapor in the western coast of Korea using atmospheric stable water isotope measurements and backward trajectory model during winter. The authors find that isotopic compositions of water vapor are strongly related with climate parameters, such as air temperature and humidity. The moisture at the study location is also strongly influenced by a mixing between condensation and evaporation processes. Moreover, the authors concluded that the moisture at the study location come from two different locations during warm and cold winter period, which are from near the north Pole and China continent, respectively. The topic of the paper is of great interest for the journal readers. The findings are also interesting for readers working on the atmospheric topic. While the work and findings are interesting, the paper needs to be improved, especially in the abstract and result sections. Conclusion does not describe all the important findings of this paper. I provide my comments below and would ask the authors to take these comments into consideration as they revise the paper.

Answer: Thank you for the positive comments and implications of our work. We have revised the abstract and conclusion as the previous two reviewers suggested.

Abstract section needs some improvements. A reason(s), why this study is important, is not described in the opening sentences. The statement saying there are few studies using water vapor isotopes is not correct. Infact there are some studies using water vapor isotopes both from the models, satellites, and in situ measurements. The main and important finding about the different moisture sources during warm and cold winter is not well stated in the abstract.

Answer: We somewhat agree with the reviewer’s point. The abstract has been revised as the reviewer suggested.

The result section needs some improvement in terms of English writing style.

Answer: The manuscript has been revised as this reviewer suggested.

The use of Iso-GSM model is not stated in the abstract as well as in the method section. The authors at least need to mention the use of Iso-GSM model for comparison in the method section and refer to Kei Yoshimura paper for detailed information about Iso-GSM.

Answer: We have revised the abstract and section of results and discussion as this reviewer suggested. In particular, we add some explanation for Iso-GSM comparison with our observed data.

As I stated in the assessment section above, the main finding about two difference moisture sources during warm and cold winter is not stated in the conclusion. The conclusion from the comparison with the Iso-GSM model is also not stated. The closing sentence from the conclusion section is confusing. Out of nowhere you hope that the data can be used for paleoclimatic studies.

Answer: We add some implications of our work in the introduction.

Please fix the equation numbers order.

Answer: We are sorry for that. We have fixed in the revised manuscript.

L11: The authors wrote: Previous studies have studied”. I would use another word to avoid similar word used twice in a sentence (study). For example: Previous studies have correlated isotopic composition of precipitation in the water cycle with climate variables.

Answer: Changed.

L12: You may write: a few studies with an article “a”. Few studies mean almost none, while a few studies mean small number but more than two. However, I do not agree with the statements that only a few studies used water vapor isotopes. There are some (see e.g., Tremoy et al., 2012; Brown et al., 2013; Risi et al., 2013; Samuels-Crow et al., 2014; Sutanto et al., 2015).

Answer: We somewhat agree with the reviewer’s point. Around Korean peninsula, what we meant was no study of water vapor isotope. There are a few studies for moisture transport using precipitation isotopes, but our study is the first study using water vapor transport with isotopes.

L13: Please rephrase the sentence. The authors said: “those isotope compositions”. It is not clear the word those refer to? Isotopes in water can be 17O as well. You need to mention δD and δ18O first.

Answer: Fixed.

L15: The authors may write “variables” since they are temperature and humidity.

Answer: Fixed.

L17-19: The sentence here is confusing. Please rephrase.

Answer: Fixed.

L19-20: The closing statement in the abstract is different with conclusion (paleoclimate).

Answer: We add a couple of lines for the linkage between our work and paleocliamte.

L26: The authors may write: Stable water isotopes.

Answer: Fixed.

L26-27: Please replace the word change in this sentence since it is used two times.

Answer: The correction has been made.

L37: Reference Dansgaard (1964) must be written as reference number.

Answer: Fixed.

L43: Here the authors stated Siberian high pressure. This is a reason why I prefer to say Siberian instead of north Pole for cold winter moisture source (see L267).

Answer: We changed this in the line L317

L49: The authors’ write: [19] described. It must be written as: Ref. [19] described. Please do the same for the rests.

Answer: Fixed.

L53: Missing “be”

Answer: Fixed.

L56: The authors may write: “….condensation temperature and saturated specific….:

Answer: Fixed.

L59: The authors may write: “…and the saturated water vapor, respectively.

Answer: Fixed.

L60: Missing comma. “….mixing ratio, and T…..”

Answer: Fixed.

L70: The authors may write: The outline of this paper.

Answer: Fixed.

L79: Superscript th.

Answer: Fixed.

L91: The authors may write: “…the analysis of precipitation isotopic composition…..”

Answer: Fixed.

L111: The authors may write: (hereafter refers to AWS). Also remove “to”.

Answer: Fixed.

L122: Here the authors mentioned about one laboratory-made standard. Please mention the name of the standard here although later the authors wrote VSMOW2, GISP, and SLAP2.

Answer: We have made the modifications accordingly.

L124: It might be: “were measured for calibration”

Answer: Fixed.

L139: Typo.

Answer: We apologize. Fixed.

L144: The authors may write: “The total precipitation of 2015/16 winter season was 77.9 mm,….

Answer: Fixed.

L145: In principle we can measure the isotopic composition of precipitation. Special care and effort are needed to avoid evaporation from the samples.

Answer: We agree with what the reviewer mentioned. However, during our study period, the amount of precipitation wasÈ so small that we were not able to obtain for isotopic analysis.

L147-148: The authors can remove the words: “that of” from the sentence. The use of respectively here is also misplaced.

Answer: We have made the modifications accordingly.

L155-156: You may write: “….depleted precipitation isotopic compositions in summer occur due to an isotope amount effect,…”

Answer: Fixed.

L157-158: The sentence is unclear. Please rephrase. May the difference between this study and others be due to the absent of precipitation in this study?

Answer: We rephrased the sentence.

L158: isotope -> isotopic.

Answer: Fixed.

L160: How about humidity? The authors only mentioned temperature. Also isotope -> isotopic. Always write isotopic composition instead of isotope composition.

Answer: The correlation with absolute humidity is not mentioned because it is described in the Mixing curves and Rayleigh distillation model.

L162-163: This sentence is confusing. Please rephrase the sentence.

Answer: Fixed.

L164-166: The authors can simply say: Like temperature, the absolute humidity also exhibits similar variability, which indicated low amount of WV during the cold surge season.

Answer: Fixed.

L166: The authors can simply say: This might be the result of isotopic fractionation.

Answer: Fixed.

L171-172: Figure 1. Misused of respectively words.

Answer: Fixed.

L177: positive -> positively.

Answer: Fixed.

L178: I am confused with two correlations values of 0.87 and 0.88.

Answer: Those are two water isotopes correlation coefficients.

L178-179: The sentence is confusing. Please rephrase.

Answer: Fixed.

L180-182: The authors may write: “This temperature effect, which is observed in precipitation isotopic compositions, can also be found in water vapor isotopic compositions”.

Answer: Fixed.

L182: The authors may replace hence with however.

Answer: Fixed.

L188: The use of word respectively here is correct.

Answer: Fixed.

L203: The word non-critical is unclear.

Answer: Fixed.

L211: I suggest to write the temperature values for warm and cold periods.

Answer: We add the temperature values with isotope values.

L218: The authors may write the equation numbers taken from the introduction section for the Rayleigh and mixing model.

Answer: We are sorry for that. It is fixed now.

L220: Double of.

Answer: Fixed.

L222: Could the authors write the reference?

Answer: We add some citations here.

L223: correlation -> correlations, variable -> variables.

Answer: Fixed.

L225: The relationships of?

Answer: Fixed.

L226: I think the sentence started from Because…… should be combined with the previous sentence.

Answer: Fixed.

L228: Remove (Figure 3). Double.

Answer: Fixed.

L228-230: The sentence is confusing and I can only see grey dot-line instead of line. Remove dot in qxδD vs q

Answer: Fixed.

L234: curve -> curves.

Answer: Fixed.

L242: Figure 3. Same lines for the last two lines (green and grey).

Answer: Fixed.

L243: Figure 3. Please mention the diamond purples are measurements. (b) q δD -> (b) qxδD.

Answer: Since we change the figure 3 to figure 4 in this version of manuscript, the reviewer’s suggestion is fixed.

L244: curves are -> curve is since it is only for Rayleigh curve. Comma before which.

Answer: Fixed.

L245: Please write comma after (25°âˆ†C). q δD/q -> qxδD/q.

Answer: Fixed.

L255: What do the authors mean with ocean inflow? Is it ocean influence?

Answer: We corrected the word. L259: The authors may write: “In addition with precipitation……”

Answer: Fixed. L261: What do the authors mean with set site?

Answer: Fixed.

L262: Moisture sources of the study period.

Answer: Fixed.

L264: Each ten period. Each should be flowed by singular.

Answer: Fixed.

L267: I think it is still a bit far away from North Pole. Why don’t the authors just say Siberia?

Answer: Fixed.

L267-269: Figure 4b and 4c are reversed.

Answer: Fixed.

L276: Where is the black line? I cannot see it. For Figure 4c, I suggest to use the same background map.

Answer: The background map of the figure 5 is provided by HYSPLIT and we can't control it. Also we have corrected misrepresentation of "black line".

L280: The authors don’t mention this model in the abstract. The authors can just simply say Iso-GSM model since they only used 1 specific GCM model.

Answer: Fixed.

L280: The authors need to mention that there is no AK application was performed or just simply say: direct comparison. Maybe they can also plot the correlation values between observed and modeled δ values. Just take the values from the model in the same time when you have observation. Data assimilation can also be used to improve the model.

Answer: We somewhat agree with the reviewer’s point.

L284: The use of word predictions here is unclear.

Answer: Fixed.

L289: It will be interesting to see if the model also produces different air mass sources during warm and cold periods. Is it possible to do that? Maybe the simplest thing is from wind direction analysis?

Answer: The question is very interesting, but it will be another work, which is beyond our scope. However, the Iso-GSM is using real wind field (nudged), such that the model somewhat represents the isotopic variations by changing of wind field.

L297: Figure 5. In the legend, please write δD observed and δ18O observed.

Answer: Fixed.

L299: typo for vapor. The word “that of” can be removed.

Answer: Fixed.

L303: The authors may write: “……moisture sources at xxx depending…”

Answer: This sentence is moved to the section of conclusion.

L305: Better always write water vapor instead of only vapor.

Answer: Fixed.

L308: The authors may replace as with shown by.

Answer: Fixed.

L309: extinguish for describing -> divert from.

Answer: Fixed.

L311-312: Please rephrase the sentence.

Answer: Fixed.

L320-321: Not a good closing sentence. In this study the authors did not mention about paleo study at all.

Answer: We add a couple of lines for a linkage between our work and paleo-climate.

Round 2

Reviewer 1 Report

The authors have addressed the reviewer's comments resulting in a much-improved manuscript. In my opinion, the article is now ready for publication. 

Reviewer 2 Report

it's ok